# Measurement of a Vibration on a Robotic Vehicle

**DOI:** 10.3390/s22228649

**Published:** 2022-11-09

**Authors:** Frantisek Klimenda, Roman Cizek, Marcin Suszynski

**Affiliations:** 1Faculty of Mechanical Engineering, University of Jan Evangelista Purkyne in Ustí nad Labem, Pasteurova 1, 400 96 Ustí nad Labem, Czech Republic; 2Faculty of Mechanical Engineering, Poznan University of Technology, 60-965 Poznan, Poland

**Keywords:** robotic vehicle, accumulators, electric motors, steering, vibration

## Abstract

This article deals with the design and construction of a robotic vehicle. The first part of the paper focuses on the selection of suitable variants for the robotic vehicle arrangement, i.e., frame, electric motors with gearboxes, wheels, steering and accumulators. Based on the selection of individual components, the robotic vehicle was built. An important part of the robotic vehicle was the design of the suspension of the front wheels. The resulting shape of the springs was experimentally developed from several design variants and subsequently produced by an additive manufacturing process. The last part of article is devoted to the experimental measurement of the acceleration transfer to the upper part of the frame during the passage of the robotic vehicle over differently arranged obstacles. Experimental measurements measured the accelerations that are transferred to the top of the robotic vehicle frame when the front wheels of the vehicle cross over the obstacle (obstacles). The maximum acceleration values are 0.0588 m/s^2^ in the *x*-axis, 0.0149 m/s^2^ in the *y*-axis and 0.5755 m/s^2^ in the *z*-axis. This experimental solution verifies the stiffness of the designed frame and the damping effect of the selected material of the designed springs on the front wheels of the robotic vehicle.

## 1. Introduction

Nowadays, we are increasingly encountering the term “robotic vehicle”. It is a machine that performs activities with a certain degree of autonomy. The vehicle performs activities at different levels of need to interact with the outside world and with the operator. A robotic vehicle moves by rolling wheels or tracks on the road surface. The vehicle is able to sense its surroundings using sensors (ultrasonic, infrared, etc.). A robotic vehicle can be controlled manually (the operator controls the vehicle manually) or autonomously (the vehicle moves along a designated path) [1,2,3,4,5].

Today, these mobile robots are widely used in various sectors of human activity, where they facilitate and often replace routine or less important work. They are used not only in the military, in rescue operations or in the exploration of unknown places, but also in companies (e.g., autonomous robotic vehicles) or in ordinary households (e.g., robotic vacuum cleaners or lawn mowers). Due to the ever-increasing number of these devices and their many different applications, it is necessary to continuously improve the control systems of these robots to achieve the desired results as efficiently as possible [6,7,8,9].

In the context of Industry 4.0, so-called AGVs (Automated Guided Vehicle) are used instead of manual or inflexible mechanical solutions (e.g., forklifts, conveyors, etc.). These trucks help to increase productivity and reduce costs associated with an internal logistics system and ensure efficient material flow. When implementing AGVs, it is important to pay attention to fleet coordination. The vehicle routing system is responsible for calculating trajectories that minimize the total distance traveled by AGVs, taking into account various constraints such as the carrying capacity of each vehicle and the plant layout where the vehicles can move, while ensuring collision-free routes [10,11,12].

When the vehicles are moving on the road, a mutual force effect is generated between the vehicle and the road. The force effects have a decisive influence on the stability, safety and driving dynamics of the vehicles. The driving dynamics include a number of parameters, especially rolling resistance, air resistance, acceleration resistance and climbing resistance. The sum of the individual resistances gives the total driving resistance that the vehicles must overcome when moving. This total resistance is equal to the minimum driving force of the drive. The amount of the rolling resistance depends on the total weight of the vehicle and the rolling resistance coefficient, i.e., mainly on the road surface. The air resistance consists of shape, swirl and frictional resistance and depends mainly on the velocity of the vehicle. The climbing resistance depends not only on the weight of the vehicle, but also on the angle of the road. When towing a trailer, the towing vehicle has to overcome the overall resistance of the trailer, which is very important because the required driving force has to be increased by it. All these driving resistances are very important in the design of vehicles (internal combustion engine, hybrid, electric, LPG (liquefied petroleum gas), CNG (compressed natural gas), hydrogen powered, etc.) [12,13,14].

Currently, mobile robotic vehicles are either without suspension or have suspension consisting of a system of springs and shock absorbers [15,16,17,18]. In this article, we will try to design the suspension based on a classic leaf spring.

The main objective of the research is divided into two sub-objectives. The first objective of the research is the design of the mobile robotic electric vehicle and its implementation. The second objective of the research is to experimentally measure the transfer of acceleration to the upper part of the robotic vehicle frame. The acceleration is caused by the robotic vehicle crossing the obstacle (obstacles). At the same time, whether the designed springs of the front wheels of the robotic vehicle can absorb part of the acceleration caused by the vehicle crossing the obstacle is experimentally measured (obstacles).

This article deals with the complete design of the electrically powered robotic vehicle for transporting small loads. Furthermore, the results of a pilot experimental measurement of acceleration transfer to the frame of the robotic vehicle during the obstacle overcoming in several configuration variations are presented. The aim is to find out what acceleration is transferred to the upper part of the frame of the designed robotic vehicle. This research serves as the basis for the design of a fully-suspended robotic vehicle that could navigate in outdoor conditions with uneven road surfaces.

In the next part of the experimental design, we will look at the battery dynamics and motor control performance. Depending on the track profile, the range of the robotic vehicle will be measured with the voltage consumption of both electric motors.

## 2. Proposal of Solution Variants

This chapter describes the design options for the mobile electric robotic vehicle. The design and selection of the suitable solution option is the first part of the first research objective.

The basic idea of the solution was to design the robotic vehicle, which will be used to transport small materials in the laboratory and at the same time to teach programming to students.

All variants of the robotic vehicle design are based on the assumption that the maximum external dimensions of the vehicle are 700 × 450 × 250 mm. The rear wheels are driven by electric motors and the front wheels are rotatable. The accumulator must provide continuous operation for 8 h. The load capacity of the vehicle is approximately 45 kg and the total weight of the vehicle must not exceed 90 kg. Another condition is a maximum vehicle velocity of 5 km·h^−1^ and a maximum road gradient of 5°.

Figure 1 shows schematically three variants of the vehicle design. The first variant of the robotic vehicle design (Figure 1a) is driven by two rear wheels. Each rear wheel is driven by its own gearbox and electric motor. The front wheel of the vehicle is rotatable by 360°. The vehicle is controlled by a control unit which, together with the electric motors, is powered by rechargeable batteries located in the vehicle’s center of gravity. The safety of the vehicle is ensured by ultrasonic distance sensors located around the perimeter of the vehicle. The advantage of this design variant is its simplicity. The disadvantages of this design concept are the stability of the vehicle when cornering, when there is a risk of overturning, and the more complicated control due to the two independent electric motors. The second version of the robotic vehicle design (Figure 1b) is structurally similar to conventional automobiles, where torque and speed are transmitted via a shaft to a differential gearbox to the drive wheels. The front wheels are rotated into travel by a servo drive with steering gear. The batteries are housed in pull-out boxes located symmetrically in the longitudinal axis of the vehicle. The control unit here controls the servo drive for the front-wheel steering and the rear-wheel drive. This control unit also evaluates sensor signals for safe vehicle movement. The advantage of this design variant is a better weight distribution of the individual vehicle components. The disadvantages of this design variant are the complex drive mechanism and the increased vehicle weight due to the weight of the drive mechanism and differential gear. The third variant of the robotic vehicle design (Figure 1c) is based on the first variant of the robotic vehicle design. In this version of the robotic vehicle design, a second swivel wheel is added at the front of the vehicle. The addition of a second turning wheel at the front of the vehicle ensures the stability of the vehicle when turning. The other parts of the vehicle are the same as in the first version of the robotic vehicle design. The advantage of this design variant is the driving stability when turning the vehicle. The disadvantage of this design variant is again the complexity of controlling two independent electric motors.

## 3. Detailed Solution of the Selected Variant of the Design of the Robotic Vehicle

This chapter is devoted to the description and implementation of the robotic vehicle according to the selected third design variant. This fulfils the first objective of the research, i.e., the design and implementation of the mobile electric robotic vehicle.

From the variants presented in the previous chapter, variant III of the proposal is selected. This solution option is selected on the basis of advantages and disadvantages. The basic components that make up the vehicle are described below.

### 3.1. Frame

The vehicle frame consists of ALUTEC kk 30 duralumin profiles, which are bolted together using corner elements. The external dimensions of the vehicle are 700 × 450 × 250 mm. The ALUTEC profiles form not only the basic perimeter frame, but also the partitions. The gearboxes with motors, the gearbox with steering, the accumulator housings, the bearing housings of the shafts with drive wheels and the swivel front wheels are bolted to the frame with crossbars.

The proposed frame was subjected to modal analysis to determine the eigenmodes and eigenvalues. Eigenmodes and eigenvalues are used to investigate the vibrations and behavior of mechanical structures, and to diagnose building or machine structures, which is one of the basic methods of dynamics. The principle of the modal analysis is based on the possibility of decomposing the oscillatory motion into partial (also modal, proper) parts, whose superposition produces the resulting motion. Each part of the oscillating motion is characterized by its own frequency, its own waveform and its own damping. By determining these resulting modal properties of the individual parts, a complete dynamic description of the mechanical system can be obtained [14]. A mechanical system with one degree of freedom (1 DOF) is shown in Figure 2.

The equation of motion of a system with one degree of freedom can be written in the form
(1)$$m\ddot{x}(t)+b\dot{x}(t)+\mathrm{kx}(t)=F(t),$$
where *m*—mass [kg], *b*—damping [Ns/m], *k*—stiffness [N/m], *x*—deflection [m], $\dot{x}$—velocity [m/s], $\ddot{x}$—acceleration [m/s^2^], *F*—excitation force [N].

If we do not consider damping, the equation of motion (1) changes to the form
(2)$$m\ddot{x}\left(t)+\mathrm{kx}(t)=F(t\right).$$

To solve the above differential equations of motion (1) and (2), we introduce initial conditions of the form
(3)$$x(0)=x_{0},\;\;\;\;\dot{x}(0)={\dot{x}}_{0}$$

If we substitute *F*(*t*) = 0, then Equation (2) will be of the form
(4)$$\ddot{x}(t)+\frac{k}{m}x\left(t\right)=0,$$
where
(5)$$\Omega_{0}^{2}=\frac{k}{m}.$$

The solution of Equation (4) is then in the form
(6)$${x(t)=C\sin(\Omega }_{0}t+\varphi ),$$
where *C*—amplitude [m], *Ω*—natural angular frequency [rad/s], *φ*—phase angle [rad].

The amplitude of the oscillation is in the form
(7)$$C=\sqrt{{\left(\frac{{\dot{x}}_{0}}{\Omega_{0}}\right)}^{2}}+x_{0}^{2}$$
and the phase angle is in the form
(8)$$\varphi =\mathrm{arctg}\frac{x_{0}\Omega_{0}}{{\dot{x}}_{0}}$$

The system thus oscillates with an angular frequency *Ω*_0_. This angular frequency is called the natural frequency of the system. Each natural frequency of the system corresponds to one natural form of oscillation. A system has as many natural frequencies as it has degrees of freedom. Equation (4) can also be written in matrix form
(9)$$M\ddot{x}(t)+\mathrm{Kx}(t)=0,$$
where **M**—mass matrix [kg], **K**—stiffness matrix [N/m], **x**(*t*)—displacement vector [m], $\ddot{x}$(*t*)—acceleration vector [m/s^2^].

The displacement is in the form
(10)$$x(t)=ye^{i\omega t}$$
and the acceleration is in the form
(11)$$\ddot{x}{(t)=-\Omega }^{2}ye^{i\omega t},$$
where **y**—own vector of the system.

By substituting Equations (10) and (11) into the motion equation in the matrix form (9) and after adjustment, we get Equation (12) in the form
(12)$$(-\Omega^{2}My+Ky)e^{i\omega t}=0$$.


After modifying Equation (12) and substituting Ω^2^ = *λ*, we get the equation in the form
(13)(K - λM)y=0.

Equation (13) is referred to as a generalized eigenvalue problem where *λ* represents the eigenvalue of the system.

The eigenvalues of the system (*λ*) are determined for the non-trivial solution of the system. Each eigenvalue *λ* has the eigenvector **y**. The eigenvalues sought must satisfyEquation (14)
(14)det|K - λM|=0.

One of the methods used to solve for natural frequencies and eigenmodes of oscillations is the Finite Element Method (FEM). To solve the modal analysis of the frame in this case, we used the ANSYS program, type Modal analysis Toolbox. The input model for this solution is the frame assembly made of ALUTEC kk profiles with a square cross-section of 30 × 30 mm. The maximum external dimensions of the frame are given at the beginning of this subchapter. SHELL181 elements were used to form the mesh, with a maximum size of 0.01 mm. The number of network elements was 859,484. The first boundary condition of the solution was tight joints at the wheel mounting points on the lower frame profiles. The second boundary condition of the solution was the compressive loading of the upper frame profiles. The total amount of force in the compressive load was 445.95 N. This amount of force is determined by the useful weight that the vehicle carries and is equal to 45 kg. The material properties of the aluminum profiles are shown in Table 1, which is 6061 aluminum alloy. The assembled frame is shown in Figure 3.

The first ten natural frequencies are shown in Table 2. The first ten natural shapes of the frame deformation are shown on Figure 4. It is clear from the individual shapes that as the number of shapes increases, so does the natural frequency of the given shape.

### 3.2. Electric Motors and Gearboxes

The robotic vehicle is powered by a two DC electric motors. The power required to drive the vehicle is determined from the sum of driving, rolling and air resistance for a maximum vehicle velocity of 5 km∙h^−1^. The air density for calculation is *ρ* = 1.25 kg∙m^−3^ (for air pressure 101 kPa and air temperature 25 °C). The size of the front impact surface is determined from the frontal projection of the vehicle, which has dimensions of 250 × 450 mm. The front impact surface has the size *S_x_* = 0.1125 m^2^. The coefficient of air resistance is chosen from the range of 0.8–1.0, as the shape of the vehicle approaches the shape of a flatbed. For the calculation, we choose the value *c_x_* = 0.81. The maximum weight of the vehicle for the calculation is *m* = 70 kg, where the weight of the load (45 kg) is included. The rise and fall of the vehicle are assumed to be a maximum of 5°. The coefficient of rolling resistance chosen for the pavement is *f* = 0.025. The power of the electric motor is given by the equation [14]. The selected parameters for calculating the power of electric motors are summarized in Table 3.
(15)$${P=(F}_{\mathrm{oval}}{+F}_{\mathrm{ovzd}}{+F}_{\mathrm{os}})\cdot \;v,$$
(16)$$P=\left(m\;\cdot \;g\;\cdot {\;f}_{\;}+\frac{1}{2}\cdot \;\rho \;\cdot {\;v}^{2}\cdot {\;c}_{x}\cdot {\;S}_{x}+G\cdot \sin\alpha \right)\cdot v,$$
(17)$$P=(70\cdot 9.81\cdot 0.025+\frac{1}{2}\cdot 1.25\cdot {\left(\frac{5}{3.6}\right)}^{2}\cdot 0.81\cdot 0.1125+70\cdot \sin\;5)\cdot \frac{5}{3.6},$$
(18)P=100.404 W,
where *P*—electric motor power (W), *F_oval_*—rolling resistance (N), *F_ovzd_*—aerodynamic air resistance (N), *F_os_*—climb resistance (N), *v*—velocity of the vehicle (m·s^−1^), *m*—maximum vehicle weight (kg), *c_x_*—air resistance coefficient (1), *S_x_*—front impact area of the vehicle (m^2^), *f*—rolling resistance coefficient (1), *ρ*—air density (kg·m^−3^), α—angle of inclination of the road (°).

Two DC electric motors, GR 53 × 58, are used to drive the vehicle, which operate at a voltage of 24 V. At this voltage of 24 V, the electric motor has a nominal power of 58.7 W, a speed of 3300 rpm, a torque moment of 0.17 Nm and a current of 19 A. The weight of one electric motor is 1.16 kg.

Attached to the GR 53 × 58 electric motors are two single-stage planetary gearboxes marked PLG 52 H. The efficiency of one gearbox is 90%. The gear ratio of the gearbox is 1:8. The permissible load in the axial/radial direction of the gearbox is 500/350 N. The continuous torque of the gearbox is 1.2 Nm. The gearbox shaft is mounted on two double ball bearings. Gears have straight teeth. A pulley is located at the output of the transmission, which is used to drive the drive wheels via V-belts.

### 3.3. Accumulators

The accumulators are used to power the electric motors of the robotic vehicle. A calculation consisting of the driving resistance (rolling, air and acceleration) will be used for the design of accumulators. The accumulators should have 24 V and their capacity should be able to power the electric motors for 8 h. Considering the velocity of the vehicle, which is 5 km∙h^−1^, the estimated driving distance is 40 km. The track for accelerating the velocity is approximately 8 km. The efficiency of the accumulators is 75%. The approximate acceleration of the vehicle is 1.389 m∙s^−2^. The coefficient of rotating masses is *δ_a_* = 1.1. More information on the coefficient of rotating masses can be found in the literature [14].

The selected parameters for calculating the energy for driving the vehicle are summarized in Table 4.

The calculation of the energy for driving the vehicle is given by the relation in the form
(19)$${E=(F}_{\mathrm{oval}}{+F}_{\mathrm{ovzd}})\cdot s_{v}{+F}_{\mathrm{ozr}}\cdot s_{r},$$
(20)$$E=(m\;\cdot \;g\;\cdot {\;f}_{\;}+\frac{1}{2}\cdot \;\rho \;\cdot {\;v}^{2}\cdot {\;c}_{x}\cdot {\;S}_{x})\;\cdot {\;s}_{v}+(m\;\cdot {\;\delta }_{a}\;\cdot \;a)\;\cdot {\;s}_{r},$$
(21)$$E=(70\cdot 9.81\cdot 0.025+\frac{1}{2}\cdot 1.18\cdot (\frac{5}{3.6}{)}^{2}\cdot 0.81\cdot 0.1125)\cdot 40,000+(70\cdot 1.1\cdot 1.389)\cdot 10,000$$
(22) E=860,021.508 J.

If 1 J = 2.778·10^−7^ kWh, then
(23)$$E=1,104,462.028\cdot 2.778\cdot 10^{-7}=0.239\;\mathrm{kWh}$$
(24)$$\;E=\frac{0.306}{\eta }=\frac{0.306}{0.75}=0.319\;\mathrm{kWh},$$
(25)$$\;E=C\;\cdot \;U=>\;C=\frac{E\;}{U}=\frac{409}{24}=13,273\;\mathrm{Ah},$$
where *E*—energy (J), *s_r_*—path for vehicle acceleration (m), *s_v_*—distance traveled by the vehicle (m), *g*—gravitational acceleration (m·s^−2^), *m*—maximum vehicle weight (kg), *δ_a_*—coefficient of rotating masses (1), *a*—vehicle acceleration (m·s^−2^).

For this battery capacity, a series-parallel connection of a Li-ION batteries marked 18,650 will be used (Figure 5a). This battery has a voltage of 3.6 V and a capacity of 2000 mAh. The battery assembled in this way will have a voltage of 25.9 V and a capacity of 18 Ah. A protection circuit and a balancer 10S 15 A will serve as protection against overcharging and discharging (Figure 5b). This balancer synchronizes the individual voltages on the parallel branches to avoid overcharging and high discharge of the individual parallel branches. The weight of the accumulator without accessories will be 2709 g.

### 3.4. Wheel Storage

The wheels used in the design of the construction of the robotic vehicle are in two versions. The front wheels are rotation. The wheels were chosen with regard to the condition that was defined at the beginning, and the maximum weight of the stroller must not exceed 70 kg. For this reason, wheels with a maximum load capacity of 60 kg per wheel were chosen.

These front wheels are bolted to the designed springs. The springs are designed as leaf springs, consisting of a main spring and two leaves below it (Figure 6a). A photo of one spring with a wheel bolted to the frame is shown in Figure 6b. The spring is screwed firmly to the carriage frame on one side and screwed into the groove on the other side so that the spring can move when the spring is deflected. The resulting shape of the springs is the 5th variation of the design. The springs were tested experimentally as the vehicle passed over unevenness and this shape of the spring is suitable for the time being because of the performed experimental measurements.

The rear wheels of the robotic vehicle are for driving. First of all, it is necessary to say how the torque is transmitted from electric motors with gearboxes to the drive wheels. The torque between the electric motor and the drive wheels is transmitted using a toothed belt transmission. The drive pulley is attached to the output shaft from the gearbox with an electric motor using a pin. The driven pulley is attached with a key to the graduated shaft and secured with a screw against displacement (Figure 7). The drive wheels are made of cast iron with a rubber cover and are equipped with steel inserts that serve to accommodate the shafts. The load capacity of one wheel is 850 kg. The diameter of the wheel is 125 mm. The shaft is connected to the wheel hub using a pin. The width of the wheel tread/hub is 50/60 mm. The vehicle’s drive wheels are mounted on shafts that are mounted in a two UCP 202 bearing housings.

The designed shaft is stressed for bending due to the gravitational force of the vehicle and for torsion transmitted from the electric motor with the gearbox to the shaft with the drive wheel. The designed shaft was checked for bending and torsion by calculation. To check the shaft for bending, the method of cuts in critical places was used. The shaft load and individual sections are shown on Figure 8. The magnitude of the resulting loading force acting at the place where the wheel is placed, the force of tension of the belt and the resulting magnitudes of the reactions at the places where they are placed were calculated from the boundary conditions of the entry and from the values given by the manufacturer.

When checking the torsional stress of the shaft, the torque is calculated and the smallest cross-section is checked. The torque is calculated for the power of the electric motor *P* = 100.404 W and the calculated speed of the driving wheel from the speed of the electric motor and the gear ratio is *n* = 6.45 s^−1^. The efficiency of the gearbox is *η_p_* = 90%. The torque is given by the relation in the form
(26)$$M_{k}=\frac{P\cdot \eta_{p}}{2\cdot \pi \cdot n}=\frac{100.404\cdot 0.9}{2\cdot \pi \cdot 6.45}=2.227\;\mathrm{Nm},$$
where *M_k_*—torque (Nm), *P*—electric motor power (W), *η_p_*—efficiency of the gearbox (1), *n*—drive wheel rotation (s^−1^).

The results of the calculated values of bending stresses and torsional stresses in individual sections are shown in Table 5.

The shafts are made of 1.0060 steel, which has a permissible bending stress of *σ_DO_* in the range of 85–115 MPa and a permissible torsional stress of *τ_DK_* in the range of 50–70 MPa. All values of bending and twisting stresses COMPLY with the permitted stresses. Each shaft is stored in the two CPU 202 bearing houses. For bearings stored in houses, the manufacturer states the dynamic load capacity *C_D_* = 7.36 kN and the static load capacity *C*_0_ = 4.48 kN. The moment condition was used to determine the reaction forces acting on the bearings in the control calculations of the shaft for bending. These are reactions *R_A_* = 220.041 N and *R_B_* = 386.659 N.

The equivalent dynamic load of the bearing is calculated using the equation in the form
(27)$$F_{e}=X\cdot F_{r}+Y\cdot F_{a},$$
where *F_e_*—equivalent dynamic load (N), *X*—coefficient of radial forces (1), *Y*—coefficient of axial forces (1), *F_r_*—radial force acting on the bearing (N), *F_a_*—axial force acting on the bearing (N).

The basic durability of the bearing is calculated using the equation in the form
(28)$$L_{10}={\left(\frac{C_{D}}{F_{e}}\right)}^{P_{B}},$$
where *L*_10_—basic bearing life (rot.), *C_D_*—dynamic bearing capacity (kN), *F_e_*—equivalent dynamic load (N), *P_B_*—exponent for ball bearings (1).

Bearing life in hours is calculated using the equation in the form
(29)$$L_{10h}={\left(\frac{C_{D}}{F_{e}}\right)}^{P_{B}}\cdot \;\left(\frac{\pi \cdot D}{1000}\right),$$
where *L*_10*h*_—bearing life in operating hours (hr), *C_D_*—dynamic bearing capacity (kN), *F_e_*—equivalent dynamic load (N), *P_B_*—exponent for ball bearings (1), *n*—drive wheel rotation (s^−1^).

The calculated bearing life values are shown in Table 6.

The service life of the bearings is a very important parameter for the overall service life of the robotic vehicle. It be seen from Table 6, the bearing closer to the belt drive has a lower service life than the bearing further from the belt drive. From the durability of the bearing closer to the belt drive, the service life is approximately 12,360 days, which is about 34 years, in a three-shift, eight-hour operation. This service life is sufficient with respect to the overall service life of the robotic vehicle.

### 3.5. Control Components

Currently, the control of the robotic vehicle is implemented for a manual control option. The circuit diagram of the control of the robotic vehicle is shown in Figure 9.

An Arduino MEGA control board was chosen for this robotic vehicle control application. The first possible solution was an Arduino UNO control board. However, this variant was rejected due to the number of digital and an analog inputs/outputs, which the Arduino UNO control board has few. The supply voltage of the Arduino MEGA control board is 5 V. For this reason, another element that was used for the control chain was an LM2596 step-down converter, which reduced the voltage supplied by the batteries from 24 V to 5 V. A BTS7960B module was used to control each of the two geared motors. These modules are powered by 24 V. Due to the possible heating of the control modules of electric motors with gearboxes up to a temperature of 70 °C, a 5 V fan was connected to the modules for heat dissipation. The last, very important component of the control chain is a Bluetooth module HC-06, which serves to connect the entire control of the robotic vehicle with the control application on a tablet.

## 4. Acceleration Measurement Methodology

The second aim of the research is to experimentally measure the transfer of deceleration to the upper part of the frame of the robotic vehicle and at the same time find out whether the tensioned springs on the front wheels of the robotic vehicle dampen parts of the induced accelerations.

This chapter is devoted to the presentation of the first results of the experimental measurement of the acceleration transfer to the upper part of the frame of the robotic vehicle. The accelerations are caused by passing the front wheels of the robotic vehicle over the obstacles with a height of 10 mm (Figure 10) in three arrangement variants (Figure 11). The first variant of the arrangement of the obstacles is shown on Figure 11a. This arrangement of the obstacles is symmetrical, where both wheels of the robotic vehicle will pass over the obstacles at the same time. The second variant of the arrangement of the obstacle is shown on Figure 11b. This is an asymmetric arrangement of the obstacle, where only the left front wheel of the robotic vehicle passes over the obstacle. The last, third variant of the arrangement of the obstacle is shown in Figure 11c. This is an asymmetric arrangement of the obstacle, where only the right front wheel of the robotic vehicle passes over the obstacle [19,20,21,22,23].

Only initial measurements are presented in this article. Other measurement results will be presented in the next article, which will follow on from this article. The measurement of the acceleration when the robotic vehicle passes over the obstacle (obstacles) was carried out at the velocity of the robotic vehicle of 2 km·h^−1^. The weight of the vehicle was around 45 kg. This is the ready weight of the robotic vehicle without load. The accelerations of only the front wheels of the robotic vehicle over the obstacle (obstacles) were measured. The aim of the measurements was to determine whether the designed springs on the front wheels of the robotic vehicle can dampen part of the acceleration caused by the wheels passing over the obstacle (obstacles).

The shape of the designed springs is described in Section 3.4. The designed springs were manufactured using FDM (Fused Deposition Modeling) additive technology. The springs were made and tested from several materials. The springs material was PLA (Polylactic Acid), PETG (Polyethylene Terephthalate Glycol) and ABS (Acrylonitrile Butadiene Styrene). All of these materials were not suitable for experimental accelerations measurements. The springs made of PLA and PETG materials broke when passing over the obstacle (obstacles). The springs made of ABS material were too stiff, and they did not absorb the accelerations caused when the wheels passed over the obstacle (obstacles). Finally, the springs were made from CPE HG-100 material. This material is co-polyester and is designed for high technical properties, great quality and easy printing. The material is suitable for technical components, especially for the production of functional prototypes and mechanical components. This material is characterized by excellent adhesion of individual layers in the additive manufacturing [24].

As already mentioned, only the front wheels of the robotic vehicle were measured during our measurement. This is due to the easier attachment of the wheels to the frame. After all measurements are completed, the suspension of the rear axle of the robotic vehicle will be designed. When designing the suspension of the rear axle, each drive wheel will have to be suspended, including the drive (electric motor with gearbox and belt drive).

The pilot experimental measurements of the transfer of the acceleration to the upper part of the frame of the robotic vehicle were carried out in mid-2022. The measurement was carried out according to the measuring chain in Figure 12.

The acceleration sensors were two TEDS three-axis piezoelectric accelerometers, 10 mV·m^−1^·s^−1^, type 4524-B from company Bruel & Kjaer. The measuring unit was a 6-channel PUlSE 3560-B-120 measuring unit from the company Bruel & Kjaer. The sensors were placed on the corners of the upper part of the frame of the robotic vehicle—see Figure 13.

The location of the sensor in the upper right corner of the frame of the robotic vehicle is shown in Figure 14. The symmetrical arrangement of the obstacles from the experimental measurement is shown in Figure 15.

## 5. Acceleration Measurement Results

The table of values with the first three dominant frequencies and the corresponding acceleration for both sensors when the robotic vehicle passes over all variants of the obstacles is in Figure 16.

The resulting graphs of the frequency curves in the *x*, *y* and *z* axes during a symmetrical passage over the obstacles by both front wheels of the vehicle (Figure 12a) are shown in Figure 17, Figure 18 and Figure 19. The resulting graphs of the frequency courses in the *x*, *y* and *z* axes during the asymmetric crossing over the obstacle by the left front wheel of the vehicle (Figure 12b) are shown in Figure 20, Figure 21 and Figure 22. The resulting graphs of the frequency courses in the *x*, *y* and *z* axes during the asymmetric crossing over the obstacle by the right front wheel of the vehicle (Figure 11c) are shown on Figure 23, Figure 24 and Figure 25. From the graphs in the *x*-axis for all variants of the arrangement of obstacles (Figure 17, Figure 20 and Figure 23), it is evident that when the wheels approach the obstacle, the wheels collide with the obstacle, which causes acceleration in this direction. The maximum measured acceleration in the *x*-axis is in the range of 0.0219 to 0.0588 m/s^2^. The acceleration in the *y*-axis should be manifested mainly in the asymmetrical crossing of the wheels over obstacles. When only one wheel passes over the obstacle, the frame of the vehicle tilts and this causes a deflection in the *y*-axis. From the graphs in the *y*-axis for all variants of the arrangement of the obstacles (Figure 18, Figure 21 and Figure 24), it can be seen that the acceleration of the frame in this axis is almost the same for all variants of the arrangement of the obstacles. The maximum measured acceleration in the *y*-axis is in the range of 0.0103 to 0.0150 m/s^2^. From this result of the solution, it follows that the frame does not tilt significantly when the front wheels pass over the obstacle asymmetrically. This fact is due to the good damping properties of the designed springs and the absorption of the acceleration by the frame. The maximum acceleration of the frame in all variants of crossing the wheels over the obstacle (obstacles) is in the *z*-axis. This s due to the fact that the frame must overcome a height difference of 10 mm in this direction, which is the height of the obstacles. From the graphs in the *z*-axis for all variants of the arrangement of obstacles (Figure 19, Figure 22 and Figure 25), it is evident that in this axis, there is a different acceleration of the frame for all variants of the arrangement of obstacles. When the front wheels pass symmetrically over the obstacles, the maximum measured acceleration on both sensors is approximately 0.5000 m/s^2^. When the left front wheel passes over the obstacle asymmetrically, the maximum acceleration of Sensor 1 is 0.3191 m/s^2^ and for Sensor 2, it is 0.2145 m/s^2^. When the right front wheel passes over the obstacle asymmetrically, the maximum acceleration of Sensor 1 is 0.3551 m/s^2^ and for Sensor 2 is 0.4228 m/s^2^, respectively In conclusion, it can be stated that the designed springs manufactured by the additive technology from the material CPE HG100 meet the sufficient damping effect in this test when the robotic vehicle passes over the obstacles.

For a clearer comparison of the frequency waveforms in the *x*, *y* and *z*-axis, Figure 26, Figure 27 and Figure 28 show the combined waveforms of the frequency waveforms for the symmetrical and asymmetrical arrangement of obstacles. The meanings of the abbreviations used in the figures are given in Table 7.

The advantage of the proposed suspension over current suspensions in use is its simplicity, compared to a suspension consisting of a coil spring and a shock absorber. Another advantage is its lower weight. Whether this suspension of the front wheels of the robotic vehicle is suitable for normal operation will be determined in the next stages of experiments. Maybe this suspension will eventually prove to be inappropriate, so all robot vehicles will use the classic coil spring and shock absorber suspension.

In the next phases of the experimental solutions, the accelerations transferred to the frame of the robotic vehicle will be measured when crossing the obstacle (obstacles) at different velocities (2, 3, 4 and 5 km·h^−1^) and under different symmetric and asymmetric loading of the robotic vehicle up to its maximum load capacity. This research will determine whether the designed springs will be satisfactory or not. If not, another material will be sought that can meet the defined operating conditions, i.e., the velocity and load of the robotic vehicle. After the experimental research of the front wheels, the suspension of the rear wheels of the robotic vehicle will be designed. The result should be the suspension of the entire robotic vehicle, which will be tested for overcoming more obstacles in different combinations than before. The aim of the entire project is to design such a suspension where the robotic vehicles can move safely even in outdoor conditions when transporting small materials, and where there are uneven surfaces, not only in production halls, where road surfaces are flat, but also in outdoor terrains with inclines.

## 6. Discussion

During the design of the frame of the robotic vehicle, the profiles from which the robotic vehicle is made from were firstly chosen. One of the proposals was to make a weldment from square steel profiles 30 × 30 × 1.5 mm in size. However, this option was rejected because the frame would be too heavy and the bulkheads could no longer be moved when the frame design was developed. The bulkheads would have to be cut off and welded elsewhere. For these reasons, the aluminum profiles of Alutecc kk 30 × 30 mm were finally chosen, which have sufficient rigidity and can be connected using a mounting elements. This construction suits us because the individual bulkheads were moved several times during the development of the robotic vehicle itself. We tried to find relevant sources that deal with this issue, but unfortunately there is no one involved in the construction of the robotic vehicle with such a frame. We found only the literature where the authors deal with the robotic vehicles that are not used for cargo transportation. These vehicles usually only have a base plate on which individual components are mounted.

In their work, Vachalek et al. [25] focus on the design and construction of a prototype of a universal robotic vehicle used for laboratory and study purposes. The chassis of the robot is designed with 6 mm and 2 mm thick dural aluminium plates. The perimeter of the chassis of the robot is made up of two symmetrical pieces welded together in the front and rear parts. The upper part is divided into a three removable blocks—the front section, the middle section and the rear section. The front section is used mainly by a membrane keyboard and the display. The middle piece can either be used to support a robotic arm or can be left empty. The rear section is intended for various control and signalization elements and power connectors.

*Design of an Autonomous Robotic Vehicle for Area Mapping and Remote Monitoring* is a work by Papoutsidakis et al. [26]. This article presents the study and the construction of a small vehicle which will have sensors and be controlled from a central control station and will also sent video feedback, so the operator can monitor the space. The frame of the robotic vehicle consists of a 5 mm thick duralumin plate, which has milled holes in it to which the individual components are attached.

Another important functional block in the design of the robotic vehicle was the drive of the vehicle. This function block includes the electric motor with the gearbox, the drive and the driven wheels. Already when choosing the boundary conditions for the design of the robotic vehicle, my colleagues and I agreed that the pilot design of the vehicle will only have the rear drive wheels. Again, we tried to find adequate resources that deal with this issue. However, most other authors design robotic vehicles with four drive wheels driven by servomotors without the gearboxes. The reason why they do not use the gearboxes and have the drive wheels directly attached to the servomotors is that their robotic vehicles are not used to transport small loads but serve as a laboratory for research. For this reason, they do not need to have higher transmitted torque for the driving wheels.

In Song’s [27] *Energy Efficient Drive of an Omnidirectional Mobile Robot with a Steerable Omnidirectional Wheels*, the omnidirectional mobile robot with steerable omnidirectional wheels (OMR-SOW) is presented. This robot can operate in either the omnidirectional or the differential drive mode depending on the drive conditions. In the omnidirectional mode, it has 3 DOFs in motion and a 1 DOF in steering, which can function as continuously variable transmission (CVT).

Based on [27], we get to another issue and that are the drive wheels. One of the variants is the already mentioned omnidirectional wheel(s). These wheels can move in all directions. These wheels can only be used if three or more wheels are used on the vehicle. In our case, when we have the steering wheels in front of the vehicle, which are not driving, unfortunately, they cannot be used. The idea of using these wheels will be realized in the second version of our prototype. For this variant, we would like to use four omnidirectional wheels and drive each wheel with its own electric motor. Due to the maximum weight of 90 kg, the already mentioned cast iron wheels with rubber casings were used as the drive wheels on our robotic vehicle.

The prototype of our robotic vehicle is currently in the second phase of implementation, where it will be equipped with sensors. These are an infrared sensors for tracking the line, and have ultrasonic distance sensors and a webcam. The first stage of development, which is described here, was the complete design of the robotic vehicle, including the drive and control. The development phases of the construction of the robotic vehicle show that, currently, the vehicle can only be controlled manually (by the operator). Upon completion of the second phase, the robotic vehicle will be able to move autonomously along the line and will be equipped with the distance sensors necessary for the safe operation of the autonomous robotic vehicles to avoid collision with a dead or live obstacle [28].

## 7. Conclusions

The first part of the article deals with the conceptual design of the robotic vehicle. A total of three conceptual layouts of the robotic vehicle were proposed, which are based on predetermined boundary conditions. Based on the advantages and disadvantages of individual conceptual designs and after agreement with colleagues about who participated in the realization of the robotic vehicle, the design variant of the robotic vehicle number III was chosen. This selected variant was further dealt with in detail. First of all, the design of the frame of the robotic vehicle was addressed, which must be rigid and assembled at the same time. The stiffness of the frame was tested using modal analysis in the ANSYS program. In the next step of implementation, the battery capacity was calculated and suitable electric motors with gearboxes were selected. Transmission of torque moment from electric motors with gearboxes to the drive wheels is ensured by toothed pulleys with a toothed belt with a gear ratio of 1:1. The control unit here is the Arduino MEGA motherboard, on which the created control system is uploaded. Connecting the vehicle to the control panel (tablet, PC) is done via Bluetooth. A control application was created to control the vehicle. The robotic vehicle prototype is currently in the first phase of implementation, where it is only controlled manually by the operator.

The next part of the article is devoted to the experimental measurement of acceleration transfer to the upper part of the frame when the front wheels pass over obstacles in three variants. This is the variant of a symmetrical crossing of the front wheels of the vehicle over the obstacle, crossing of the left front wheel over the obstacle and crossing of the right front wheel over the obstacle. With these experimental measurements, we verified both the rigidity of the frame and the damping effect of the designed springs produced by additive technology. The accelerations were measured in two places of the frame in the *x*, *y* and *z* axes. The result of the experimental measurements is that the designed springs are suitable for now. The experimental measurements carried out so far have shown us that the suspension designed by us meets our laboratory conditions. Therefore, there is no need to have suspension consisting of a coil spring and shock absorber as in other mobile robot vehicles. However, the research is still in its early stages and only after further experiments will it become clear whether we were right or not.

In the next phase of the research, we will focus on crossing an obstacle (obstacles) at different velocities and with different loads of the robotic vehicle. After the completion of these experiments and in order to determine whether the proposed suspension is suitable, the suspension of the rear drive wheels of the vehicle will be designed and implemented. This will suspend the entire vehicle and it will be experimentally tested again. In the final phase, the vehicle will be rebuilt, an electric motor will be added to the front wheels of the vehicle and the wheels will be replaced with other off-road wheels.

## Figures and Tables

**Figure 1 sensors-22-08649-f001:**
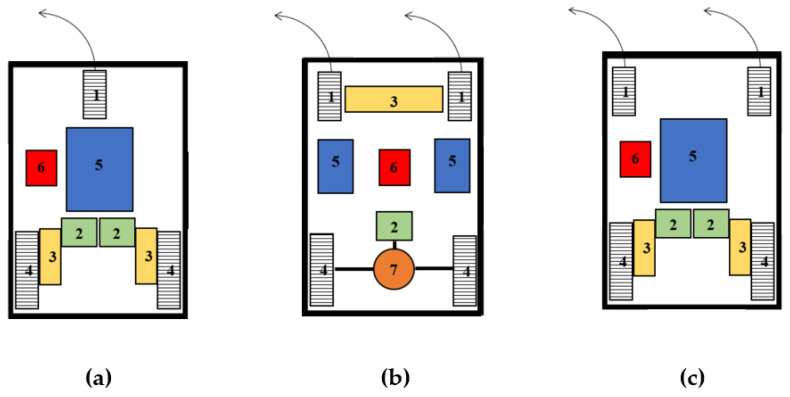
Scheme of variant I (**a**), variant II (**b**), variant III (**c**) (Legend: 1—steering wheel, 2—electric motor, 3—distribution mechanism, 4—wheel drive, 5—accumulators, 6—control unit, 7—transmission with differential).

**Figure 2 sensors-22-08649-f002:**
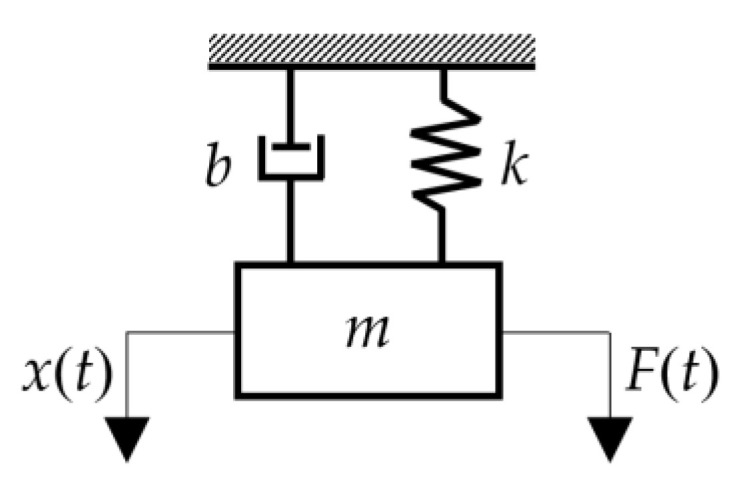
Scheme of the system with one degree of freedom (Legend: *m*—mass, *b*—damping, *k*—stiffness, *x*(*t*)—deflection, *F*(*t*)—excitation force).

**Figure 3 sensors-22-08649-f003:**
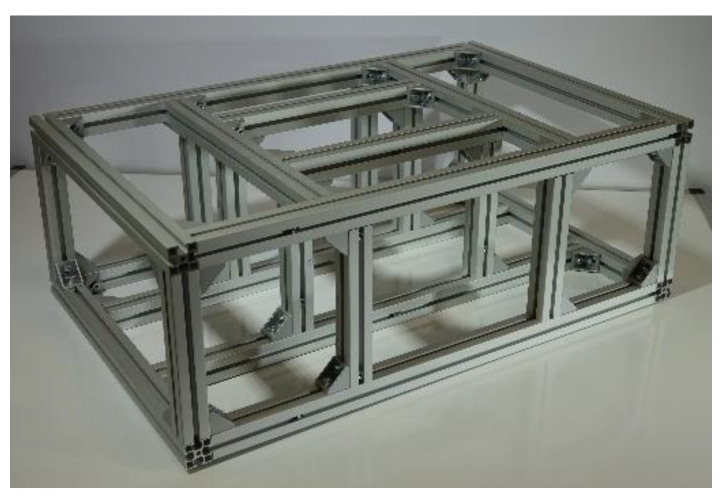
Assembled frame—photo.

**Figure 4 sensors-22-08649-f004:**
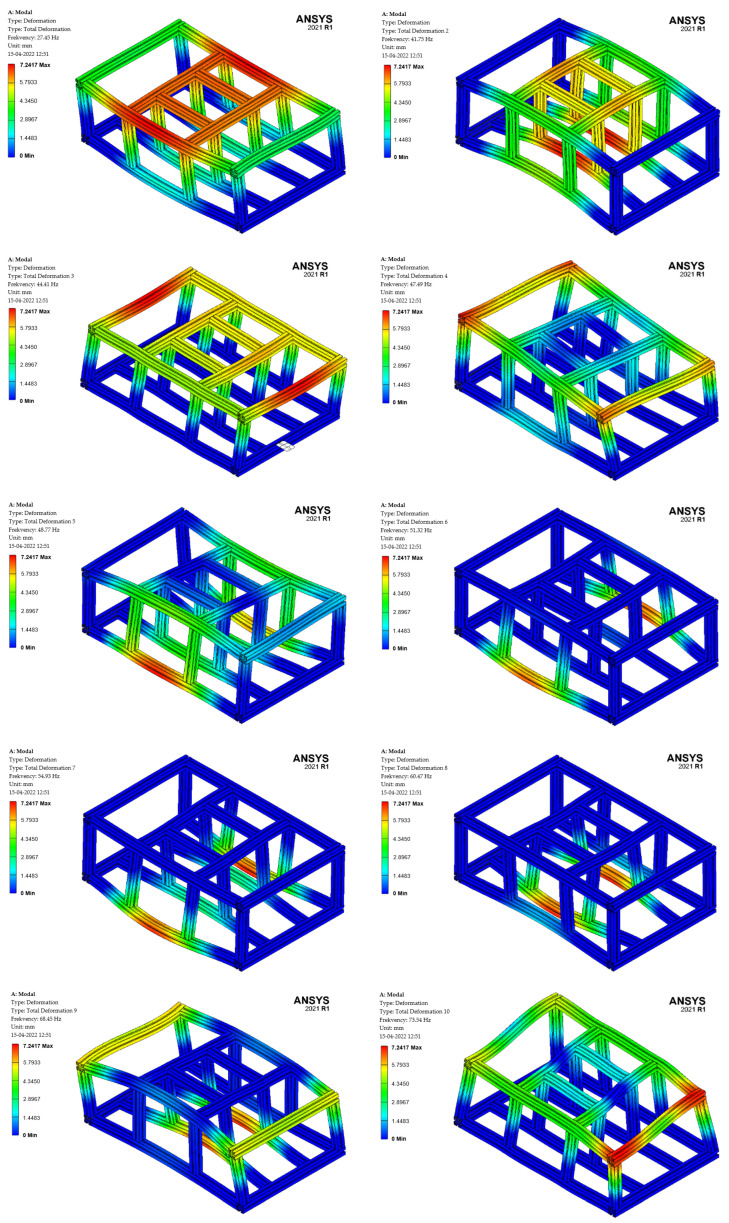
Custom shapes of frame oscillations.

**Figure 5 sensors-22-08649-f005:**
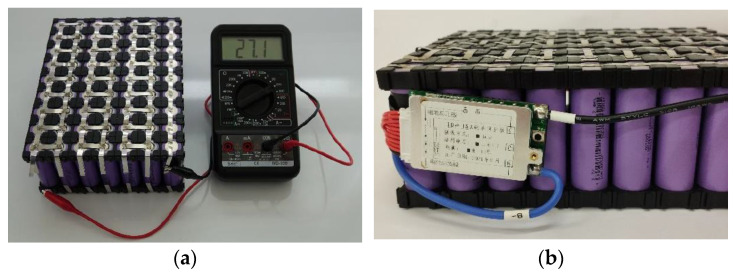
(**a**) Accumulator assembly; (**b**) Protective balancer 10S 15A.

**Figure 6 sensors-22-08649-f006:**
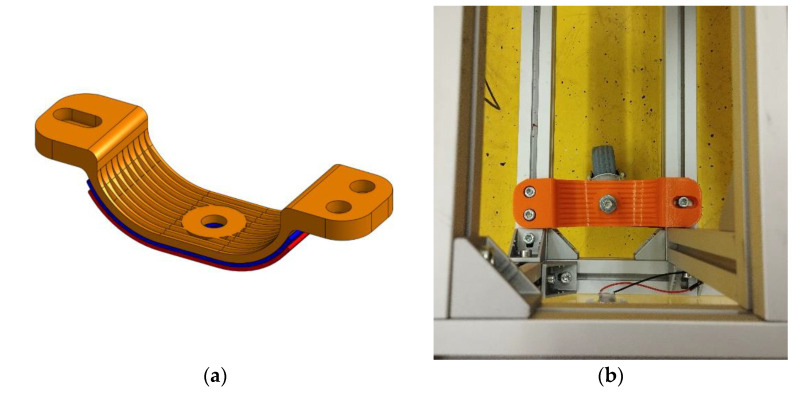
(**a**) Model of the designed spring; (**b**) Spring with wheel screwed to the frame.

**Figure 7 sensors-22-08649-f007:**
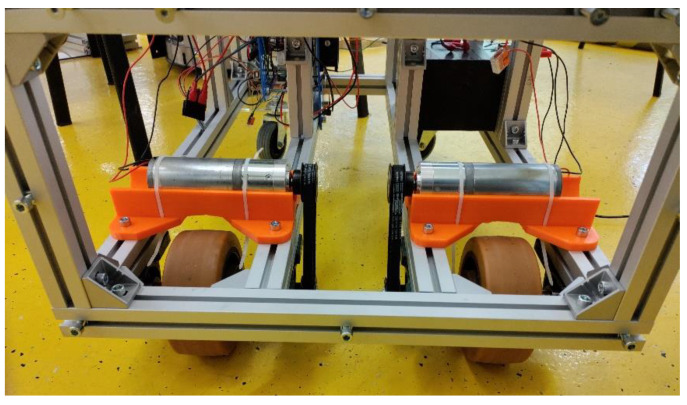
Transmission of torques from electric motors with gearboxes to drive wheels.

**Figure 8 sensors-22-08649-f008:**
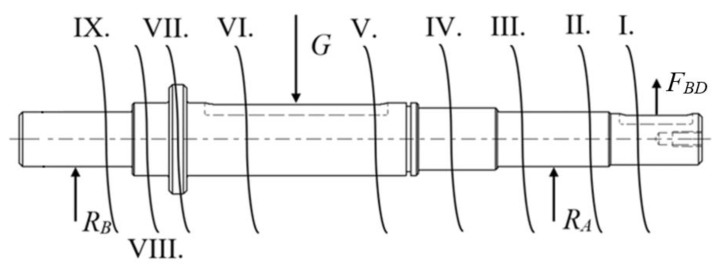
Loaded shaft with marked sections (Legend: G—heaviness, *F_BD_*—force from the belt drive, *R_A_*—reaction, *R_B_*—reaction, I.–IX.—marked sections).

**Figure 9 sensors-22-08649-f009:**
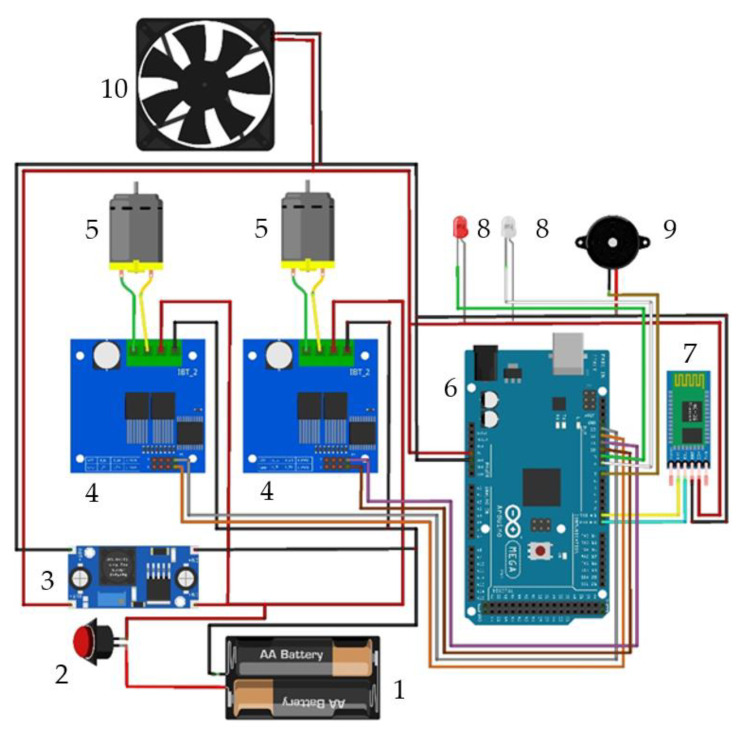
Control circuit diagram for the robotic vehicle (1—batteries, 2—on/off button, 3—LM2596 step-down converter, 4—BTS7960B module controls for motors, 5—electromotors, 6—Arduino MEGA, 7—Bluetooth module HC-06, 8—LEDs. 9—buzzer, 10—fan).

**Figure 10 sensors-22-08649-f010:**
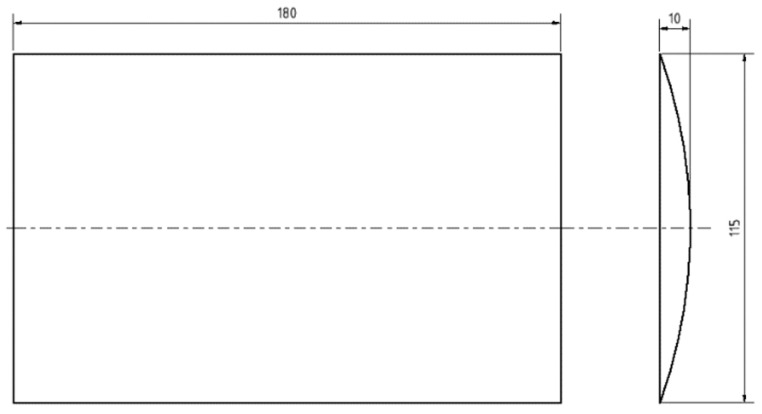
Obstacle dimensions.

**Figure 11 sensors-22-08649-f011:**
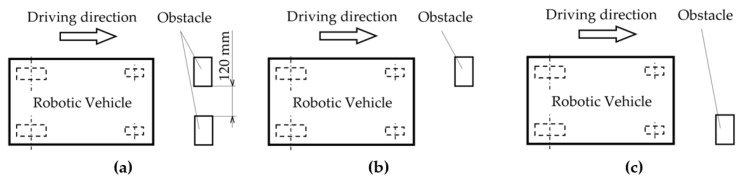
Obstacle variants; (**a**) symmetrically, (**b**) asymmetrically to the left, (**c**) asymmetrically to the right.

**Figure 12 sensors-22-08649-f012:**
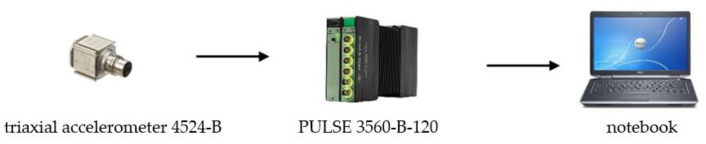
The measuring chain for measuring the acceleration.

**Figure 13 sensors-22-08649-f013:**
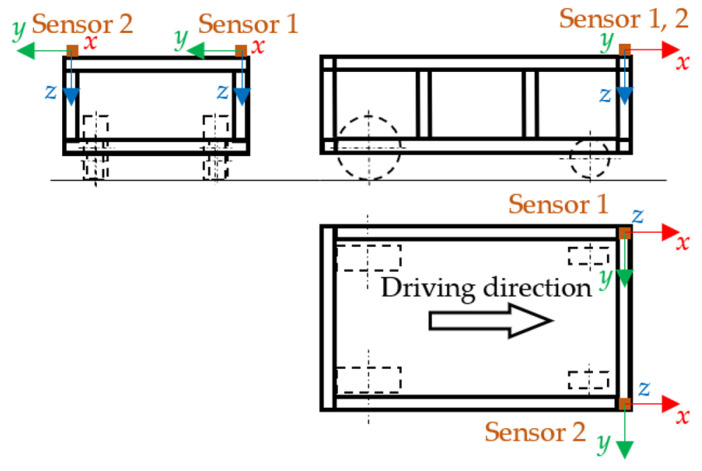
Location of the sensors on the robotic vehicle.

**Figure 14 sensors-22-08649-f014:**
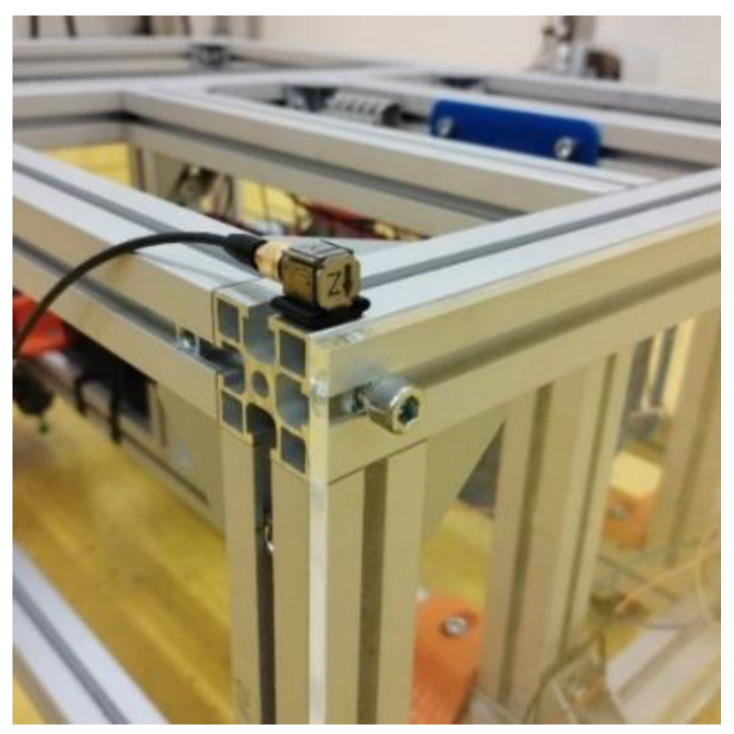
Position of the sensor in the upper right corner of the frame—photo.

**Figure 15 sensors-22-08649-f015:**
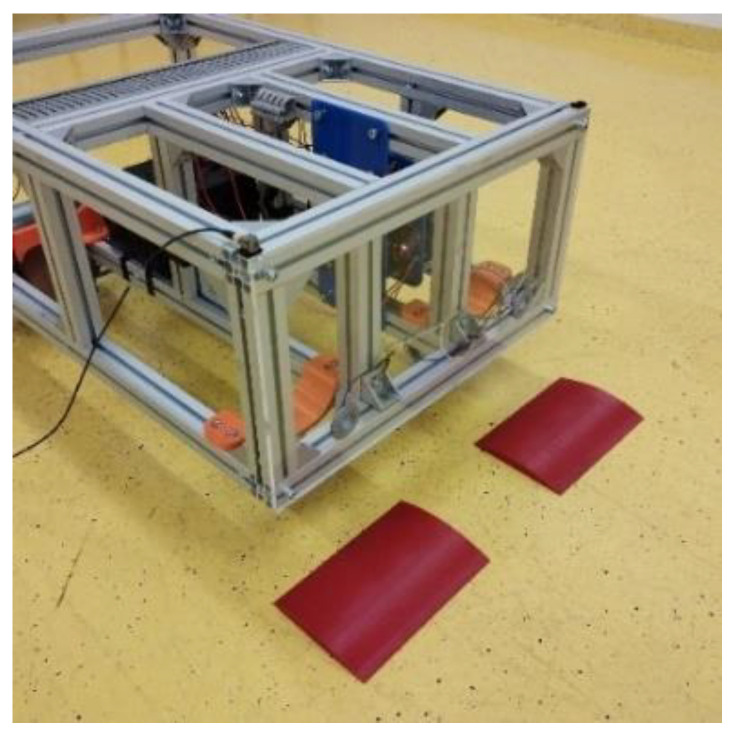
Symmetrical arrangement of the obstacles—photo.

**Figure 16 sensors-22-08649-f016:**
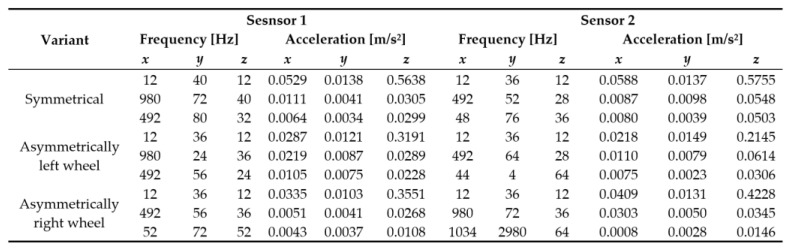
The resulting dominant frequencies and the corresponding acceleration when crossing the obstacles.

**Figure 17 sensors-22-08649-f017:**
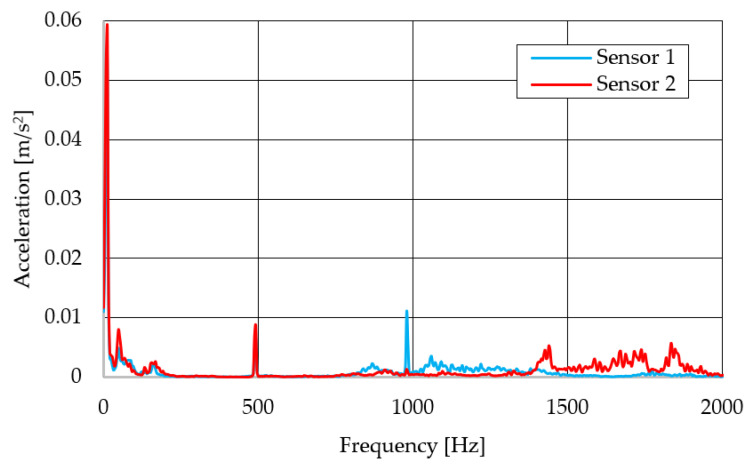
Comparison of the frequency waveforms in the *x*-axis during symmetrical crossing over the obstacles.

**Figure 18 sensors-22-08649-f018:**
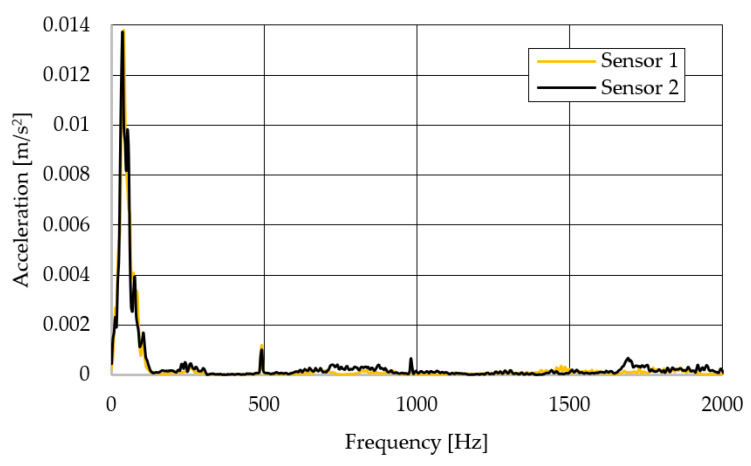
Comparison of the frequency waveforms in the *y*-axis during symmetrical crossing over the obstacles.

**Figure 19 sensors-22-08649-f019:**
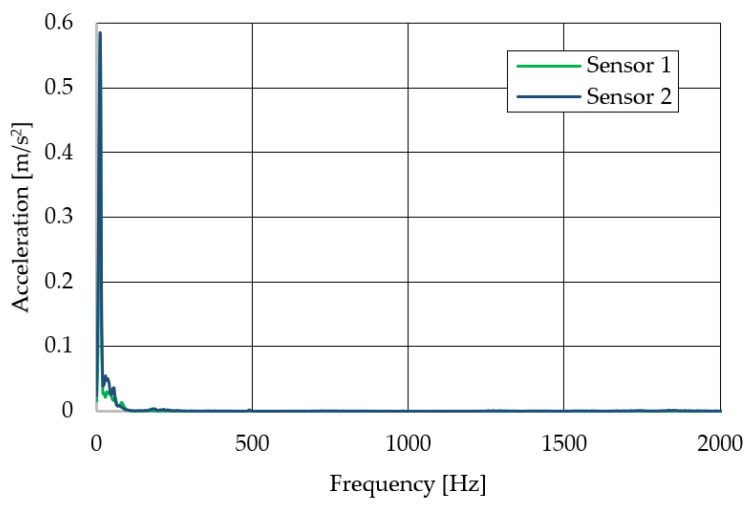
Comparison of the frequency waveforms in the *z*-axis during symmetrical crossing over the obstacles.

**Figure 20 sensors-22-08649-f020:**
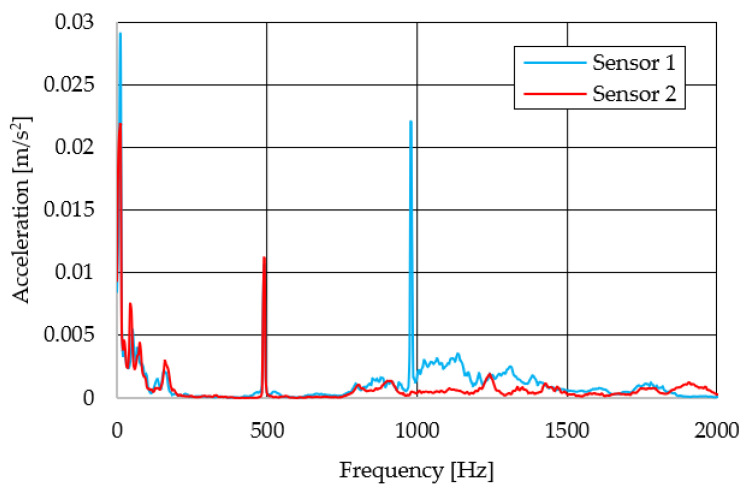
Comparison of the frequency waveforms in the *x*-axis during asymmetrical crossing over the obstacle by the left wheel.

**Figure 21 sensors-22-08649-f021:**
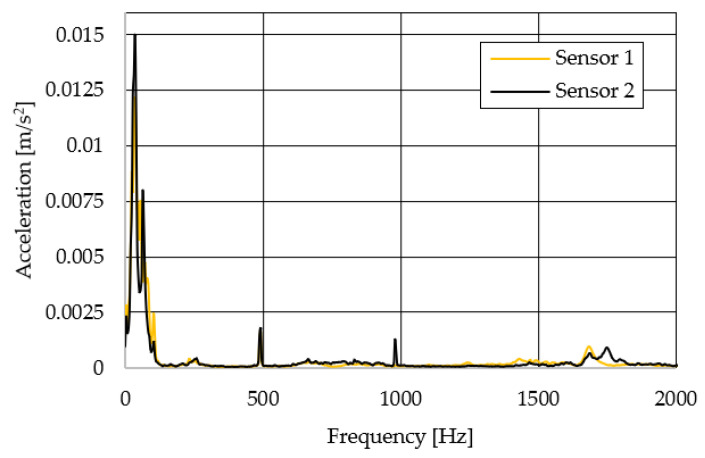
Comparison of the frequency waveforms in the *y*-axis during asymmetrical crossing over the obstacle by the left wheel.

**Figure 22 sensors-22-08649-f022:**
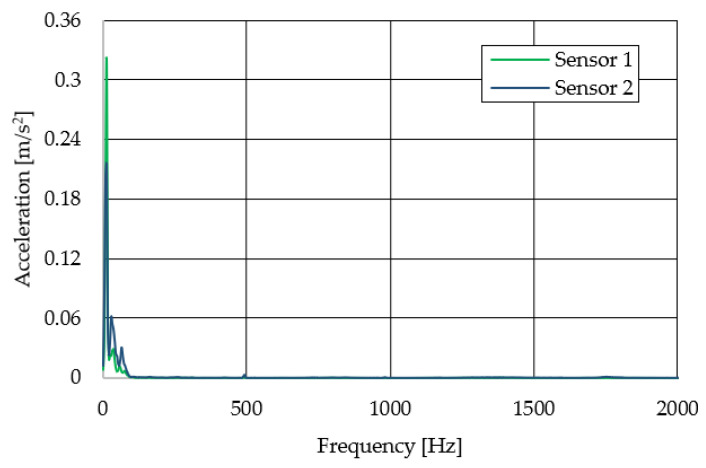
Comparison of the frequency waveforms in the *z*-axis during asymmetrical crossing over the obstacle by the left wheel.

**Figure 23 sensors-22-08649-f023:**
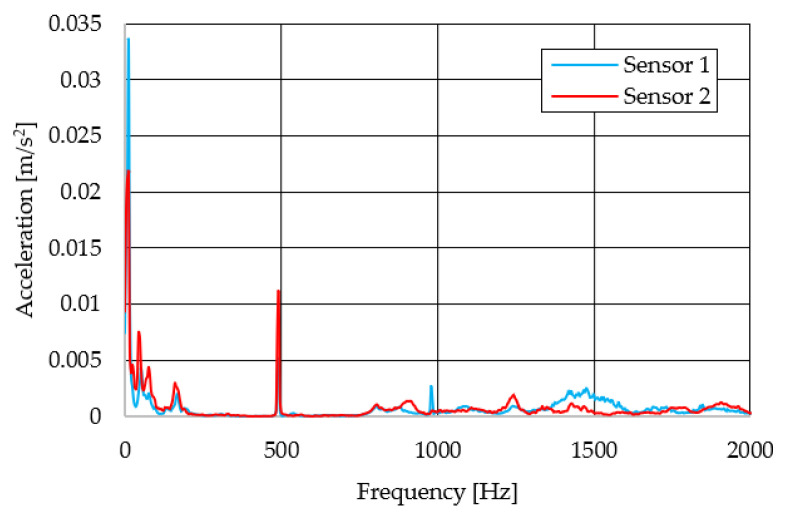
Comparison of the frequency waveforms in the *x*-axis during asymmetrical crossing over the obstacle by the right wheel.

**Figure 24 sensors-22-08649-f024:**
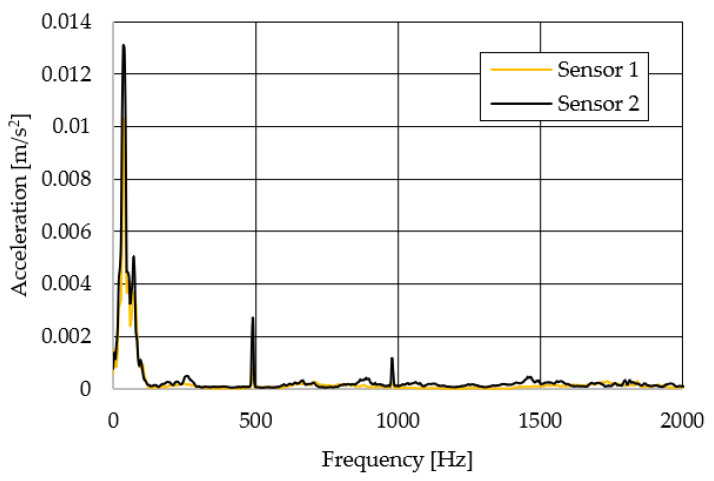
Comparison of the frequency waveforms in the *y*-axis during asymmetrical crossing over the obstacle by the right wheel.

**Figure 25 sensors-22-08649-f025:**
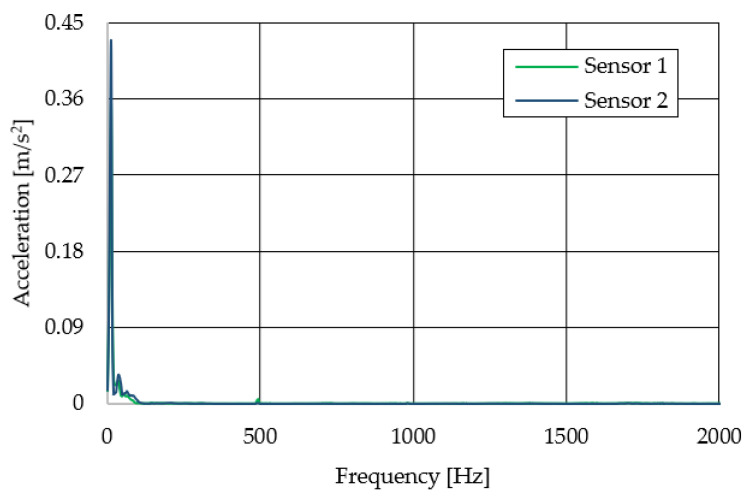
Comparison of the frequency waveforms in the *z*-axis during asymmetrical crossing over the obstacle by the right wheel.

**Figure 26 sensors-22-08649-f026:**
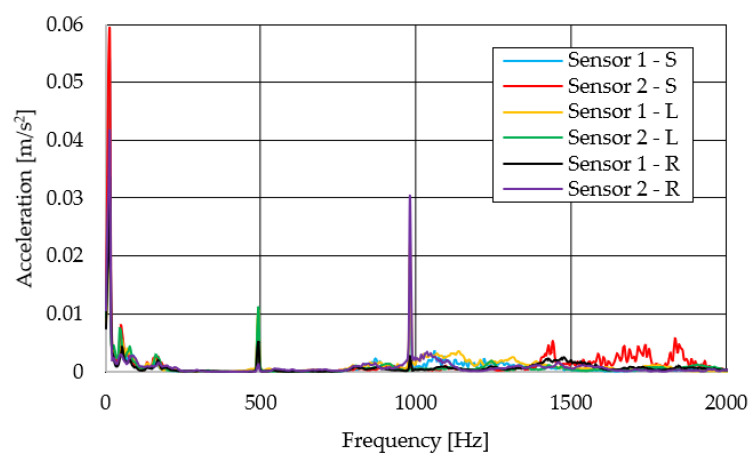
Comparison of the frequency waveforms in the *x*-axis.

**Figure 27 sensors-22-08649-f027:**
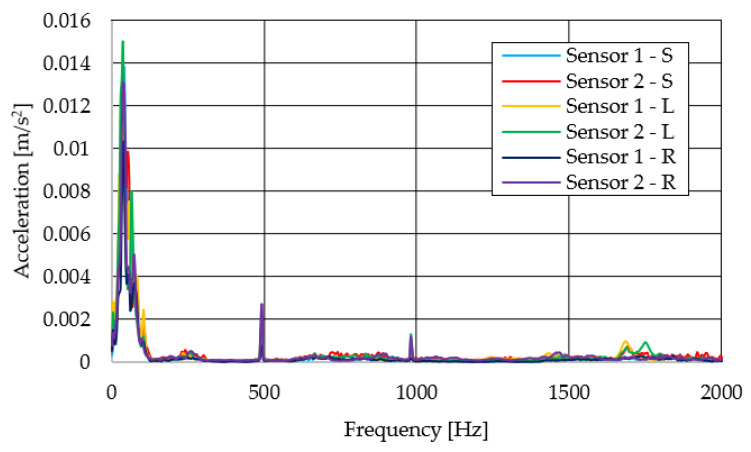
Comparison of the frequency waveforms in the *y*-axis.

**Figure 28 sensors-22-08649-f028:**
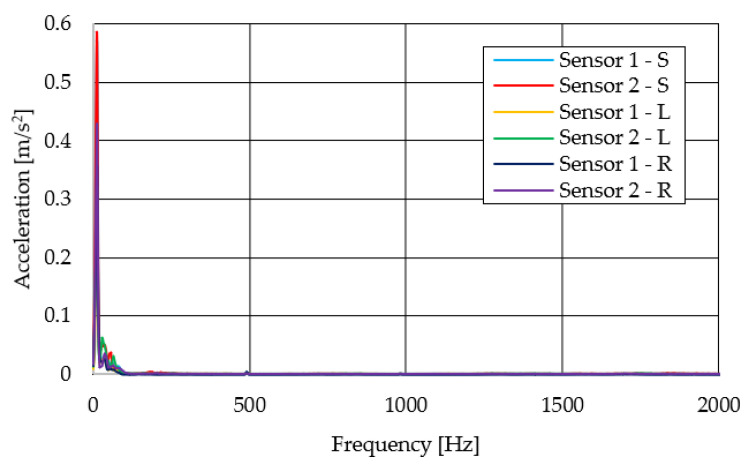
Comparison of the frequency waveforms in the *z*-axis.

**Table 1 sensors-22-08649-t001:** The material properties of the aluminum alloy 6061.

Name	Unit	Value
Modulus of elasticity in tension	Pa	6.9 × 10^10^
Poisson’s number	-	0.33
Density	kg/m^3^	2700

**Table 2 sensors-22-08649-t002:** The natural frequency of the frame.

Order	Frequency [Hz]	Order	Frequency [Hz]
1.	27.45	6.	51.32
2.	41.75	7.	54.93
3.	44.41	8.	60.47
4.	47.49	9.	68.45
5.	48.77	10.	73.54

**Table 3 sensors-22-08649-t003:** Selected parameters for calculating the power of electric motors.

Parameter	Value
vehicle velocity	5 km·h^−1^
air density	1.25 kg·m^−3^
front impact surface	0.1125 m^2^
coefficient of air resistance	0.81
maximum weight	70 kg
rise and fall of the vehicle	5°
coefficient of rolling resistance (pavement)	0.025

**Table 4 sensors-22-08649-t004:** Selected parameters for calculating the energy for driving the vehicle.

Parameter	Value
track length	8 km
efficiency of the accumulators	75%
acceleration of the vehicle	1.389 m·s^−2^
coefficient of rotating masses	1.1

**Table 5 sensors-22-08649-t005:** Calculated values of the bending stresses and torsional stresses in the individual sections.

Section	Bending Moment in the Section [Nmm]	Bending Cross-Sectional Modulus [mm^3^]	Bending Stress [MPa]	Torsion Cross Section Modulus [mm^3^]	Torsional Stress [MPa]
I.	1000.00	134.27	7.45	268.53	8.29
II.	2240.00	331.34	6.76	662.68	3.36
III.	8295.11	331.34	25.04	662.68	3.36
IV.	16,772.27	482.33	34.77	1570.80	1.42
V.	29,780.98	365.59	81.46	731.18	3.05
VI.	15,013.55	365.59	41.07	731.18	3.05
VII.	12,552.32	2650.72	4.74	5301.44	0.42
VIII.	7629.84	785.40	9.71	1570.80	1.42
IX.	0	331.34	0.00	662.68	3.36

**Table 6 sensors-22-08649-t006:** The calculated bearing life values.

Parameter	Bearing Housing *R_A_*	Bearing Housing *R_B_*
The equivalent dynamic bearing [N]	220.041	386.659
Basic bearing durability [10^6^ rot.]	37,421.704	6896.814
Bearing durability in hours [hr]	1,609,535.667	296,637.174

**Table 7 sensors-22-08649-t007:** Abbreviation meaning.

Abbreviation	Abbreviation Meaning
Sensor 1—S	Sensor 1—symmetrical crossing over the obstacles
Sensor 2—S	Sensor 2—symmetrical crossing over the obstacles
Sensor 1—L	Sensor 1—asymmetrical crossing over the obstacle by the left wheel
Sensor 2—L	Sensor 2—asymmetrical crossing over the obstacle by the left wheel
Sensor 1—R	Sensor 1—asymmetrical crossing over the obstacle by the right wheel
Sensor 2—R	Sensor 2—asymmetrical crossing over the obstacle by the right wheel

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
