# Peer review of "Measurement of a Vibration on a Robotic Vehicle"

_sensors, 2022, doi:10.3390/s22228649_

Round 1

Reviewer 1 Report

This paper presents a vibration measurement on a robotic vehicle. I would suggest publishing this paper as the quality of the work is high. I can suggest rechecking the formatting and English writing. Also, I think the introduction can improve. As the work can be very interesting for the readers, I would suggest presenting the information of the study as much as possible. 

Author Response

The answers for Reviewer 1 are in the appendix.

Reviewer 2 Report

Please find the comments, attached.

Author Response

The answers for Reviewer 2 are in the appendix.

Reviewer 3 Report

The paper seems to be dedicated to enhance the design and the construction of a vehicle prototype, but the presentation has a lot of limitations, some being mentioned below.

1. The content of the work does not correspond to the title chosen by the authors.

2. The mathematical model of a spring-mass-damper system is a very well known model and is redundant together with the Fig. 2. Moreover, the differential equation (1) is3. In the  written as for a laboratory work with the students.

3. The acronyms LPG and CNG are not defined the first time they appear in the text.

4. At 3.2, all the parameters and tehnical data should be inserted in a table otherwise it is difficut to follow the text.

5. The Figure 5 illustrates a photo with a motor, but such a picture does not make sense.

6. The Figures 18 to 24 must be ploted in the same figure and explained comparatively

7. Section 6 should be the state of the art and must be inserted in the Introduction along with the motivation of the paper. 

8. The manuscript should also be carefully arranged.

9. The exact problem, the methodology and the contributions are not clear in the first two sections. Sec. II mentions a solution to a problem that was not clearly defined.

Author Response

The answers for Reviewer 3 are in the appendix.

Round 2

Reviewer 3 Report

There are still mistakes in expression and mistakes in writing, for example "mass metrix".

I encourage the authors to write a rigorous text in which they address to all reviewers the improvements made and the corrected and introduced text, not as a draft as it is now.

Author Response

Dear Reviewer, thank you for your effort spent to revision of our article.

Throughout the article, we have had the English language checked by our English teacher. The article is already saved as a version without revisions.

Round 3

Reviewer 3 Report

The paper can be accepted in the present form.